# Mitigating the Impact of Temperature Variations on Ultrasonic Guided Wave-Based Structural Health Monitoring through Variational Autoencoders

**DOI:** 10.3390/s24051494

**Published:** 2024-02-25

**Authors:** Rafael Junges, Luca Lomazzi, Lorenzo Miele, Marco Giglio, Francesco Cadini

**Affiliations:** Politecnico di Milano, Department of Mechanical Engineering, Via La Masa n.1, 20156 Milan, Italy; rafael.junges@polimi.it (R.J.); lorenzo.miele@mail.polimi.it (L.M.); marco.giglio@polimi.it (M.G.); francesco.cadini@polimi.it (F.C.)

**Keywords:** generative artificial intelligence, variational autoencoder, temperature, ultrasonic guided wave

## Abstract

Structural health monitoring (SHM) has become paramount for developing cheaper and more reliable maintenance policies. The advantages coming from adopting such process have turned out to be particularly evident when dealing with plated structures. In this context, state-of-the-art methods are based on exciting and acquiring ultrasonic-guided waves through a permanently installed sensor network. A baseline is registered when the structure is healthy, and newly acquired signals are compared to it to detect, localize, and quantify damage. To this purpose, the performance of traditional methods has been overcome by data-driven approaches, which allow processing a larger amount of data without losing diagnostic information. However, to date, no diagnostic method can deal with varying environmental and operational conditions (EOCs). This work aims to present a proof-of-concept that state-of-the-art machine learning methods can be used for reducing the impact of EOCs on the performance of damage diagnosis methods. Generative artificial intelligence was leveraged to mitigate the impact of temperature variations on ultrasonic guided wave-based SHM. Specifically, variational autoencoders and singular value decomposition were combined to learn the influence of temperature on guided waves. After training, the generative part of the algorithm was used to reconstruct signals at new unseen temperatures. Moreover, a refined version of the algorithm called forced variational autoencoder was introduced to further improve the reconstruction capabilities. The accuracy of the proposed framework was demonstrated against real measurements on a composite plate.

## 1. Introduction

In recent years, there has been a notable surge in the exploration of structural health monitoring (SHM) based on ultrasonic guided wave propagation for damage detection, localization, and quantification. This method is specific to plate and shell-like structures and is based on the deployment of piezoelectric transducers that serve as actuators, transmitting ultrasonic waves through the material, while other similar transducers serve as receivers to capture the transmitted waves. The discernible contrast between a baseline signal recorded when the structure is intact and a signal from an unknown state of the structure may indicate the presence of damage [1,2,3]. In [4,5,6], this method was explored through numerical simulations and experimental validations to detect and characterize damage. In [7,8,9,10] these methods were further developed with the introduction of innovative approaches to enhance the accuracy and efficiency of damage identification by leveraging temporal information, physical knowledge, and the capabilities of deep learning networks. Furthermore, refs. [11,12,13] aimed to overcome challenges associated with information loss during feature extraction, biases in feature selection, limitations in composite material considerations, and the need for large high-fidelity datasets, using for instance, a combination of unsupervised data-driven methods with tomographic approaches without the need for extensive training datasets. Furthermore, it is crucial to acknowledge that changes to the waves are not exclusively indicative of structural alterations within the monitored system but may also be influenced by various environmental and operational conditions (EOCs), such as moisture, vibration, and especially temperature, as exhibited in [14,15,16].

In Abbassi et al. [17], autoencoders demonstrated the capability to detect damage at different positions independently of temperature, which varied from 20 °C to 60 °C. The training dataset encompassed data from all tested temperatures. However, it is notable that, in general, some temperatures are difficult to maintain in the laboratory environment while being very common during the operational life of the structure. For example, aircraft at cruise altitude encounter temperatures as low as −50 °C, a challenging condition to be replicated and maintained in the laboratory. Variational autoencoders (VAEs) [18,19] are a possible solution to this issue.

The VAE introduces a probabilistic interpretation of its results by modelling the latent space as a probability distribution. After training, the model can be used for generating new realistic data by sampling from the learned latent space distribution, thereby generating data samples that differ from the input but remain faithful to the underlying behaviour of the data. Notably, a VAE was previously employed in Shu et al. [20] to predict the displacement at various points on a dam, demonstrating lower prediction errors compared to traditional models such as the long short-term memory model. The VAE was capable of extracting features from all environmental data, e.g., water level, dam temperature, water temperature, and rainfall, while traditional models only extracted primary features, leaving information out of the analysis.

The novelty in this study lies in applying the VAE to predict ultrasonic guided wave signals travelling through a composite panel at temperatures not present in its training dataset. Furthermore, linearity of the latent space points was enforced by introducing a new loss function based on singular value decomposition (SVD), as it was expected that the impact of raising the temperature would be antisymmetric to the impact of lowering the temperature. A comparison between the VAE’s prediction error with and without this restriction was conducted. The aim of this work is to present a proof-of-concept that generative AI can be used to neutralize the effect of EOCs on the damage-diagnosis task. In fact, the proposed tool can be used to reconstruct signals under varying EOCs, thus enhancing the damage fingerprint and helping damage-diagnosis algorithms reduce false alarms and improve the quality of predictions.

The structure of this article is as follows: Section 2, *Materials and Methods*, provides a comprehensive description of the dataset used in this study and outlines the implementation of the VAEs. Section 3, *Results*, showcases the signal prediction outcomes for the diverse scenarios examined in this study. Section 4, *Discussion*, presents insightful comments regarding the obtained results. Finally, Section 5, *Conclusions*, offers a summarized overview of the conclusions drawn out from this study.

## 2. Materials and Methods

A generative artificial intelligence algorithm was trained over a dataset of ultrasonic guided waves acquired over different temperatures with the aim of learning temperature-related features. After training, the model was used for generating signals at new temperatures.

### 2.1. Dataset

The dataset utilized in this study was sourced from the OpenGuidedWaves platform [15], a repository offering comprehensive datasets of wave signals acquired on a carbon-fibre-reinforced polymer (CFRP) plate which has a quasi-isotropic layup with a stacking sequence of [45/0/−45/90/−45/0/45/90] S. The plate’s dimensions measure 500 mm × 500 mm, and it has a thickness of 2 mm. The experimental setup involved 12 piezoelectric sensors. Signals were acquired in a round-robin fashion, including measurements from 40 kHz to 260 kHz, with a step of 20 kHz. The sampling process was carried out under varying temperature conditions, encompassing a temperature range from 20 °C to 60 °C. Notably, temperature was varied in a cyclical manner, involving two complete cycles. As a result, a comprehensive dataset of 322 distinct temperatures was acquired.

In this work, the original dataset was split into three distinct subdatasets, designed to simulate diverse scenarios:**Standard dataset:** This dataset encompassed all the 322 samples;**Band dataset:** This dataset included all the signals acquired between 30 °C and 50 °C;**Sparse dataset:** This dataset comprised clusters of samples at nearby temperatures, strategically spaced with a fixed interval; that is, clusters with a radius of 2 °C and separated by 5 °C were considered.

This approach allowed for the creation of multiple datasets, providing varied case studies to enhance the model’s adaptability and performance across different temperature scenarios.

### 2.2. Generative Artificial Intelligence Models

#### 2.2.1. Variational Autoencoder

The model employed in this work was a VAE, an improved version of the traditional autoencoder.

An autoencoder is a neural network architecture designed for unsupervised learning. Comprising an encoder and a decoder, it aims to learn efficient representations of input data. The encoder transforms the input data into a compressed, lower-dimensional representation known as the latent space representation of the data. This encoded information is then decoded by the decoder to reconstruct the original input.

Mathematically, an autoencoder minimizes the reconstruction error, promoting the automatic learning of meaningful features in the data. The reconstruction loss shown in Equation (Equation 1), often represented as $L_{\mathrm{recon}}$, is typically defined as the mean squared error (MSE) between the input *X* and the reconstructed output $\hat{X}$:(1)$$L_{\mathrm{recon}}\left(X,\hat{X}\right)=\frac{1}{N}\sum_{i=1}^{N}{∥X_{i}-{\hat{X}}_{i}∥}^{2}$$
where *N* is the number of data points in the signals *X* and $\hat{X}$.

The ability of autoencoders to learn compact representations of data makes them valuable for tasks where extracting meaningful features is crucial. For example, autoencoders find applications in various domains, such as dimensionality reduction, feature learning, and anomaly detection. Moreover, the versatility of their architecture allows autoencoders to be tailored to specific learning objectives. Leveraging this characteristic, several variations of the standard fully connected architecture have been proposed in the literature, including denoising autoencoders and VAEs. Particularly promising capabilities characterize VAEs, which are described by a latent space characterized by probability distributions rather than single deterministic points, as illustrated in Figure 1. This enables the generation of new data points through sampling of the learned latent space distributions. Let z be a latent variable representing a sample from the latent space distributions and μ and σ be the mean and standard deviation of the latent space distributions. The encoder network parametrizes the distributions. During training, the model is encouraged to learn distributions that follow standard normal distributions according to Equation (Equation 2):(2)z=μ+σ⊙ϵ
where z is the sampled latent vector, μ is the mean vector obtained from the encoder, σ is standard deviation vector obtained from the encoder, ϵ is the random vector sampled from a standard normal distribution, and ⊙ is the element-wise multiplication operator.

The reconstruction loss ($L_{\mathrm{recon}}$) is added with the Kullback–Leibler (KL) divergence term ($L_{\mathrm{KL}}$) shown in Equation (Equation 3), which measures the difference between the learned distributions and the standard normal distributions:(3)$$L_{\mathrm{KL}}=-\frac{1}{2}\sum_{i=1}^{M}\left(1+\log\left(\sigma_{i}^{2}\right)-\mu_{i}^{2}-\sigma_{i}^{2}\right)$$
where *M* is the number of variables, i.e., distributions, considered in the latent space.

Hence, the overall VAE loss presented in Equation (Equation 4) is a combination of the reconstruction loss and the KL divergence term:(4)$$L_{\mathrm{VAE}}=L_{\mathrm{recon}}+\beta \cdot L_{\mathrm{KL}}$$
where β is a hyperparameter that controls the importance of the KL divergence term in the overall loss.

In summary, VAEs leverage probabilistic encoding to enable the generation of continuous and structured latent space representations, from which new data points can be created. The inclusion of the KL divergence term promotes the learning of well-behaved latent space distributions in such a manner that all latent space points are demoted if they are far from the centre of the latent space.

The VAE architecture considered in this work is shown in Table 1.

#### 2.2.2. Forced Variational Autoencoder

In this work, an enhanced version of the standard VAE, i.e., the forced VAE (f-VAE), is proposed. The f-VAE is an autoencoder with enforced linearity within the latent space. This is achieved through the introduction of an SVD component. The model architecture includes an encoder, a decoder, and a latent-space sampling layer, each contributing to the overall VAE structure. A novel addition to the loss function is the SVD loss term. After the entire dataset is encoded into the latent space, the mean is subtracted and SVD is performed. The SVD loss term is presented in Equation (Equation 5):(5)$$\min\left(\sum_{i=2}^{n}\frac{\sigma_{i}}{\sigma_{1}}\right)$$
where $\sigma_{i}$ is the i-th singular value, and *n* is the number of singular values computed.

Hence, the total loss in Equation (Equation 6) is a combination of the reconstruction loss, KL divergence, and the introduced SVD loss, as follows:(6)$$L_{\mathrm{VAE}}=L_{\mathrm{recon}}+\beta \cdot L_{\mathrm{KL}}+\gamma \cdot L_{\mathrm{SVD}}$$
where β and γ are hyperparameters controlling the weight of KL divergence and SVD loss, respectively.

### 2.3. Training and Signals Generation

The f-VAE and VAE models were trained on all three proposed datasets. The workflow describing how training and signal reconstruction were addressed in this work is shown in Figure 2.

The following hyperparameters were tuned to optimize the training performance:Learning rate;Batch size;Number of epochs;Kullback–Leiber loss weight;SVD loss weight.

Among the best performing values, the set of hyperparameters that allowed for a latent space characterized by a linear correspondence to the variation of the network inputs was selected.

After training, signals at target temperatures were generated for testing the generation performance of the models. The following steps were followed for generating signals:**Temperature selection:** The target temperature for signal generation was chosen. This temperature served as the basis for the desired signal.**Model initialization:** The pretrained model was initialized, including loading of the trained weights and preparation of the model for signal generation.**Latent-space interpolation:** SVD was used to elucidate the connection between the latent-space coordinates, i.e., $z_{1}$ and $z_{2}$ in this work, and temperature, with the direction of maximum variance being discerned. The primary direction was considered to fully characterize the learned trend in the latent space, enabling a unified entry point into the latent space representing the signal temperature.**Signal reconstruction:** The decoder was used to reconstruct the signal corresponding to the selected temperature.

By employing SVD to map the chosen temperature to latent-space coordinates and by subsequently utilizing the VAE’s decoder, this approach enabled the generation of signals that reflected the desired temperature. Moreover, employing SVD exclusively on the training data ensured that the interpolation line captured only the known data, simulating a real-world scenario where models are trained with limited data. As the method operates in an unsupervised manner, temperature labels were not supplied during training. Moreover, testing was performed by generating signals at temperatures not considered during training so as to verify the generalization capabilities of the proposed method.

The performance of the model during the generation phase was evaluated using the following error metrics:**Root mean square error (RMSE):** This is a measure of the average magnitude of the differences between the reconstructed signals and the original signals, which is calculated according to Equation (Equation 7).
(7)$$\mathrm{RMSE}=\sqrt{\frac{1}{N}\sum_{i=1}^{N}{\left(x_{i}-{\hat{x}}_{i}\right)}^{2}}$$
where *N* is the number of data points, $x_{i}$ is the i-th data point of the original signal, and ${\hat{x}}_{i}$ is its reconstruction.**Signals comparison:** Different signals at different temperatures were qualitatively compared to visualize if the generated signal matched the expected result.

This metrics provided insights into how well the f-VAEs and VAEs were able to reconstruct signals starting from latent-space representations. The generated signals spanned the entire temperature range in the original dataset described in [15] even though the training dataset for certain models did not encompass signals from the entire range. This intentional extension beyond the training dataset mirrored a testing scenario, allowing for a comprehensive evaluation of the models’ performance and their ability to interpolate and extrapolate out of the training set.

## 3. Results

The performance of the two models (VAE and f-VAE) was evaluated against the three different datasets described in Section 2.1. Each pairing of model and dataset was scrutinized with consideration to the latent-space linearity, RMSE, and the qualitative evaluation of the signals generated at four distinct temperatures: 25 °C, 35 °C, 45 °C, and 55 °C. Moreover, the generated signals were also compared to the signal at 40 °C, i.e., to the signal acquired at the median temperature value in the training datasets, in order to verify the interpolation and extrapolation capabilities of the proposed methods.

Without losing generality, all the considerations reported in this Section refer to latent space representations characterized by zero variance. That is, in the interest of clarity, only the mean values of the latent space distributions were considered.

### 3.1. VAE

#### 3.1.1. Standard Dataset

The distribution of the learned latent space representations and the reconstruction error related to the VAE model trained over the standard dataset are shown in Figure 3. Figure 3a reveals distinct points aligned with a clear direction of primary variance. Indeed, the SVD method underscored a discernible gap between the first and other components, endorsing the reliability of the interpolation line. Notably, the RMSE shown in Figure 3b consistently depicted an error below 4.5% universally, with the lowest reconstruction error at around 40 °C, and higher errors at the tails of the distribution. This behaviour may indicate that the model failed to accurately learn temperature-related features and always reconstructed the signal at the median temperature in the training dataset.

The qualitative comparison of the generated signals is shown in Figure 4. The results underscored the tendency of the generated signals to closely follow the signal at the median temperature in the training dataset, i.e., 40 °C, rather than adhering to the dataset signal at the corresponding temperature. This behaviour confirms the intuition that the VAE trained on the standard dataset failed to learn how guided waves are influenced by temperature.

#### 3.1.2. Band Dataset

The distribution of the learned latent-space representations and the reconstruction error related to the VAE model trained over the band dataset are shown in Figure 5. Similar observations to those already reported in Section 3.1.1 emerged. The latent-space plot shown in Figure 5a demonstrated a pronounced alignment of points along the primary variance direction, reaffirming the efficacy of the interpolation line through SVD. Due to the dataset’s limited range in temperature, the tails of the dataset extended beyond the SVD interpolation line, as the VAE was not explicitly trained on those regions. The RMSE shown in Figure 5b mirrored the trend observed for the standard dataset. That is, errors were consistently below 4.5%, the lowest error was observed at around 40 °C, and highest RMSEs characterized the distribution tails. Similarly, Figure 6 shows that the generated signals qualitatively resembled the signal at 40 °C rather than those at the target temperature.

The results described above allow us to conclude that the VAE trained over the band dataset was not able to learn temperature-related features.

#### 3.1.3. Sparse Dataset

The distribution of the learned latent space representations and the reconstruction error related to the VAE model trained over the sparse dataset are shown in Figure 7. The latent-space plot shown in Figure 7a reveals a distribution markedly different from the preceding datasets in Figure 3a and Figure 5a, where a more pronounced direction of variance was observed through SVD. In this case, instead, both the first and second components carried significance, showcasing a distinctive characteristic of the dataset. As a consequence, the encoded signals revealed a sinusoidal pattern instead of a linear trend.

Despite this potentially disadvantageous behaviour, the RMSE plot shown in Figure 7b follows a similar trend to the previous datasets, consistently emphasizing errors at the distribution tails. Interestingly, the RMSE pattern aligns with that characterizing the previously discussed datasets.

The signal reconstruction capabilities shown in Figure 8 are consistent with those of the VAEs trained on the other two datasets. The generated signals tend to closely resemble the signal at 40 °C, rather than adhering to the signal at the target temperature.

### 3.2. f-VAE

#### 3.2.1. Standard Dataset

The distribution of the learned latent-space representations and the reconstruction error related to the f-VAE model trained over the standard dataset are shown in Figure 9. Conspicuous differences compared to the VAE model presented in Section 3.1.1 can be appreciated. The latent-space distribution shown in Figure 9a exhibited a more linear trend, closely resembling a straight line. Notably, SVD underscores a significantly larger first component compared to secondary ones, implying the negligible contribution of these latter components. Also, the RMSE shown in Figure 9b presented a different behaviour than that observed for the VAE model. That is, no pronounced increase in error characterized the tails of the distribution. Except for a few points at higher temperatures approaching a 4% error, the majority of points did not exceed a 2% overall error. Remarkably, 90% of the points remained below the 1% error threshold.

The qualitative comparison of the generated signals is shown in Figure 10. The f-VAE model clearly outperformed the VAE model in terms of the accuracy of the generated signals. In fact, the generated signals exhibited a closer resemblance to the expected signals, rather than strictly adhering to the signal at 40 °C.

The results showed that forcing VAEs to learn linear representations in the latent space allowed for the influence of temperature on ultrasonic guided waves to be captured correctly.

#### 3.2.2. Band Dataset

The distribution of the learned latent-space representations and the reconstruction error related to the f-VAE model trained over the band dataset are shown in Figure 11. Distinguishable variations in comparison to the VAE model outlined in Section 3.1.2 can be observed. The latent-space distribution depicted in Figure 11a displayed a more linear tendency similar to the one described in Section 3.2.1. The RMSE illustrated in Figure 11b exhibited a pattern similar to that noted in the VAE model but with some discrepancies: the tails exhibited higher errors up to 3.5%, but the RMSE within the temperature range of 30 °C and 50 °C showed greater consistency, staying below 1%.

The generated signals presented in Figure 12, in line with the model’s behaviour observed in the standard dataset, continued to closely follow the dataset signals. The f-VAE model outperformed the VAE model in terms of accuracy of the generated signals, as already highlighted in Section 3.2.1. Despite the inherent challenges posed by extreme temperature points, the f-VAE was able to generate signals at temperatures out of the training dataset. That is, the f-VAE was also able to extrapolate.

#### 3.2.3. Sparse Dataset

The f-VAE trained over the sparse dataset was characterized by satisfactory performance, as shown in Figure 13. The model was able to capture the primary sources of variance within the sparse dataset even though the linearity was not as pronounced as that observed for the f-VAE trained over the standard dataset (Figure 9a). Major differences are observable by comparing the the f-VAE and the VAE trained over the same sparse dataset. In fact, while the latent space shown in Figure 13a plot resembled a linear behaviour, the VAE learned a sinusoidal pattern (Figure 7a) characterized by a nonnegligible second singular value.

Also, the RMSE plot shown in Figure 13b displays a satisfactory performance, consistently maintaining errors below 2.5%. Notably, 90% of the points fell below the 1% error threshold, indicating the model’s adaptation to the complexities of the sparse dataset.

In line with the observed trends in Figure 10 and Figure 12, the generated signals shown in Figure 14 faithfully followed the expected signals. Also here, the f-VAE model outperformed the VAE model in terms of accuracy of the generated signals.

## 4. Discussion

The analysis of the performance of the two models across different datasets revealed the potentialities and limitations of the employed generative artificial intelligence algorithms.

The VAE trained over the standard dataset seemed to effectively capture the relation between signal temperature and the latent space coordinates, as supported by the SVD analysis. Although the reconstruction error over the test dataset was satisfactorily low, a trend indicating that the model was only able to reproduce signals at 40 °C rather than strictly adhering to the expected signal at the target temperature was identified.

Similar considerations were drawn out from the analysis of the performance of the VAE trained over the band dataset. In fact, the latent-space distribution was aligned along a clear primary variance direction. Despite SVD capturing the maximum variance in the dataset, the bandwidth considered in the band dataset introduced challenges, resulting in high reconstruction errors at the tails of the RMSE distribution. Additionally, the generated signals still matched the 40 °C signal, regardless of the target temperature.

The same unsatisfactory generation capabilities characterized the VAE trained over the sparse dataset. Here, the performance was even worse, given that the latent-space distribution was characterized by two nonnegligible singular values.

The introduction of the f-VAE brought about notable improvements. When trained over the standard dataset, the f-VAE introduced a more linear latent space, with a significantly larger first component according to SVD. The RMSEs over the test dataset were considerably lower than those characterizing the VAEs. No clear trend of higher reconstruction errors at the tails of the temperature distribution was identified, indicating enhanced precision in signal generation. Furthermore, the generated signals closely resembled the expected signals at the target temperatures.

Similarly, the f-VAE trained over the band dataset was characterized by a linear latent-space representation in the temperature range considered during training. The regression line slightly departed from the linear trend at unseen temperatures. The same trend was shown by the reconstruction error, which was characterized by higher values when extrapolation was performed. This behaviour is commonly shown by all machine learning algorithms, which cannot be fully trusted when extrapolating. Still, the generated signals closely followed the expected signals at all temperatures, constituting evidence of the capability of the model to generate realistic signals.

The f-VAE trained over the sparse dataset offered satisfactory performance. The latent space exhibited a prominent linearity, and the RMSE was kept low at all temperatures. Accordingly, the generated signals closely matched the expected signals. The reconstruction quality achieved using the sparse dataset was higher than that characterizing the f-VAE trained over the band dataset. This indicates that while f-VAEs work better when interpolating, caution should be taken when they are extrapolating.

Higher reconstruction errors characterized the signals close to 60 °C generated by the f-VAEs. This behaviour came from the dataset composition. In fact, signals were acquired by varying temperature in a cyclic manner. By doing so, the dataset included two acquisitions at 60 °C and four acquisitions at 20 °C and throughout the rest of the dataset. This discrepancy implied a less densely populated training distribution in regions close to 60 °C.

Signal generation involved negligible computational time, in the order of milliseconds, despite training taking a few minutes on a hexacore Intel i7-10850 CPU at 2.70 GHz. Scaling up to larger structures with numerous sensors is expected to lead to longer training times since the network architecture would need to be enlarged for better capturing signal variability. However, the time for generating new signals is expected not to be affected and to remain in the order of milliseconds. The same considerations apply to training over signals acquired over a wider temperature range.

## 5. Conclusions

In this work, variational autoencoders and singular value decomposition were used to discern the influence of temperature on ultrasonic guided waves. Moreover, a newly developed machine learning algorithm, i.e., the forced variational autoencoder, was introduced to further improve the reconstruction capabilities of the generative artificial-intelligence-based framework. The accuracy of the proposed method has been demonstrated against real measurements on a composite plate. The following conclusions can be drawn:Regardless of the composition of the training dataset, traditional variational autoencoders cannot learn how to generate signals at different temperatures.Satisfactory reconstruction accuracy was demonstrated by the forced variational autoencoders coupled with singular value decomposition.Forced variational autoencoders can work in realistic scenarios, even when the training dataset is sparse.

Future work will focus on implementing forced variational autoencoders and singular value decomposition into unsupervised frameworks for damage detection, localization, and quantification. This will represent a step towards robust structural health-monitoring tools that are not influenced by varying environmental and operational conditions.

## Figures and Tables

**Figure 1 sensors-24-01494-f001:**
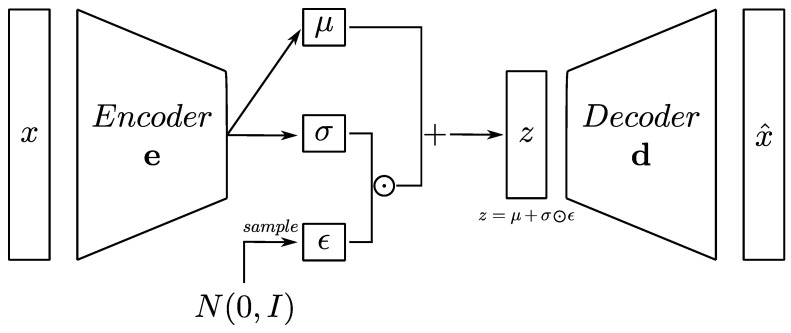
VAE Architecture.

**Figure 2 sensors-24-01494-f002:**
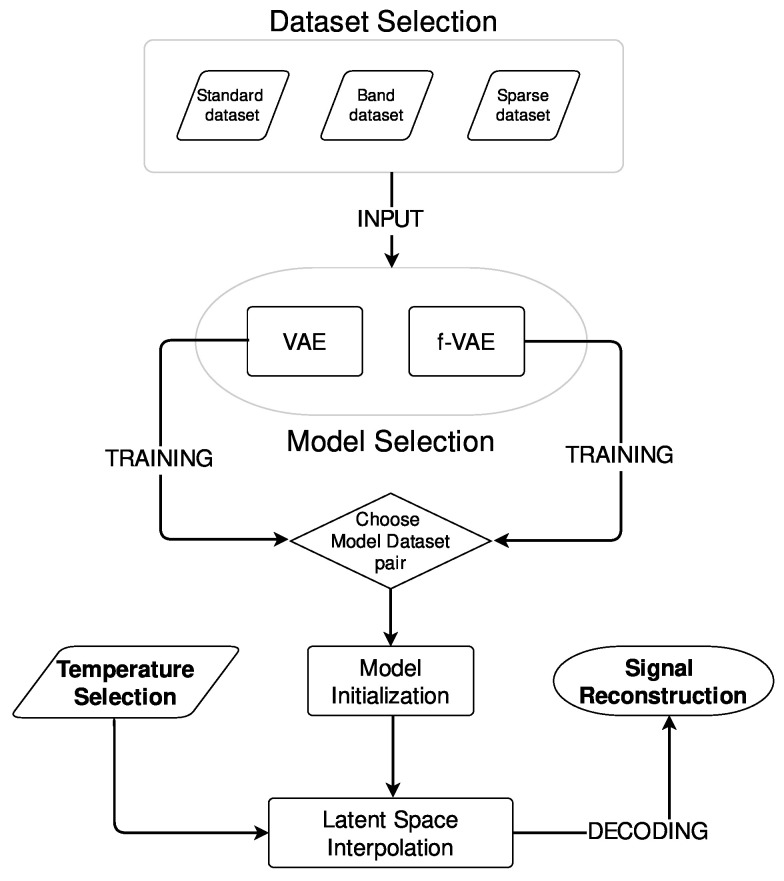
Workflow describing training and signal generation.

**Figure 3 sensors-24-01494-f003:**
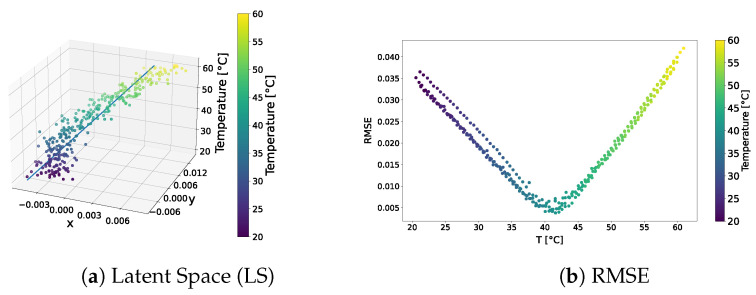
VAE—standard dataset. Latent-space distribution and RMSE.

**Figure 4 sensors-24-01494-f004:**
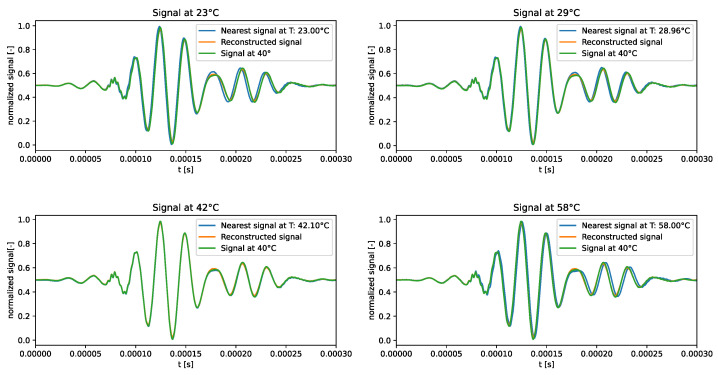
Test signals generated by the VAE trained over the standard dataset.

**Figure 5 sensors-24-01494-f005:**
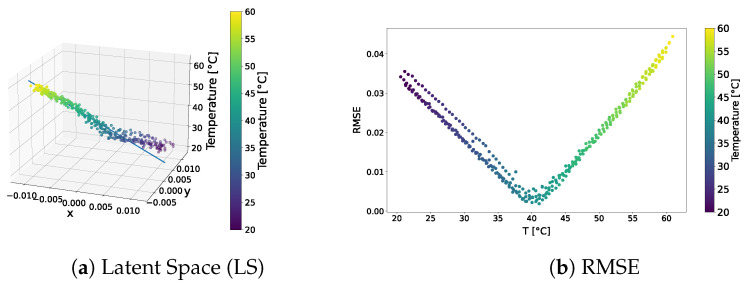
VAE–band dataset. Latent space distribution and RMSE.

**Figure 6 sensors-24-01494-f006:**
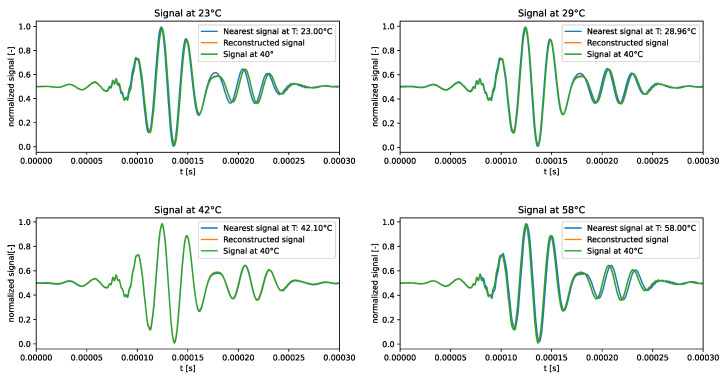
Test signals generated by the VAE trained over the band dataset.

**Figure 7 sensors-24-01494-f007:**
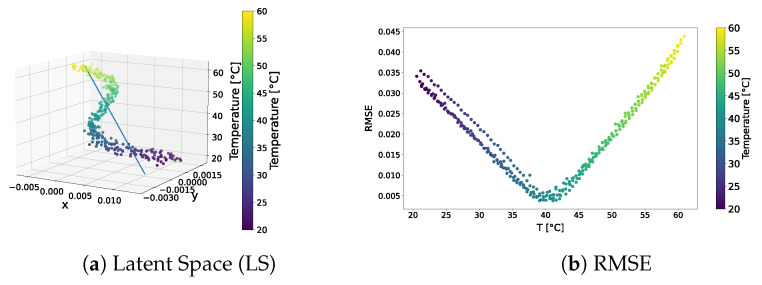
VAE—sparse dataset. Latent-space distribution and RMSE.

**Figure 8 sensors-24-01494-f008:**
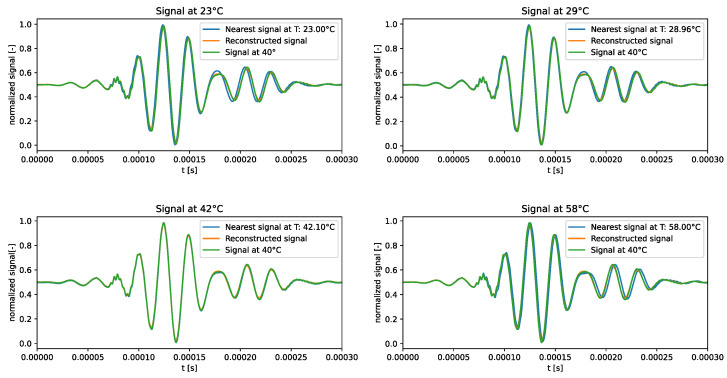
Test signals generated by the VAE trained over the sparse dataset.

**Figure 9 sensors-24-01494-f009:**
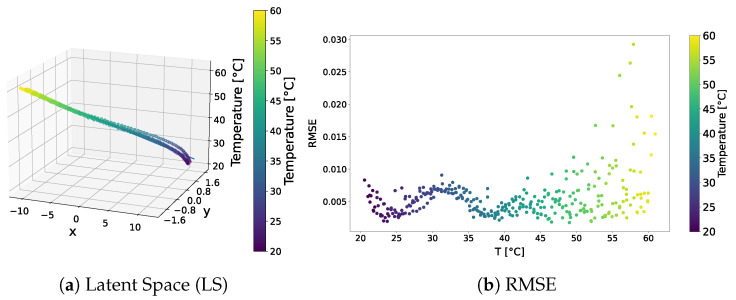
f-VAE—standard dataset. Latent-space distribution and RMSE.

**Figure 10 sensors-24-01494-f010:**
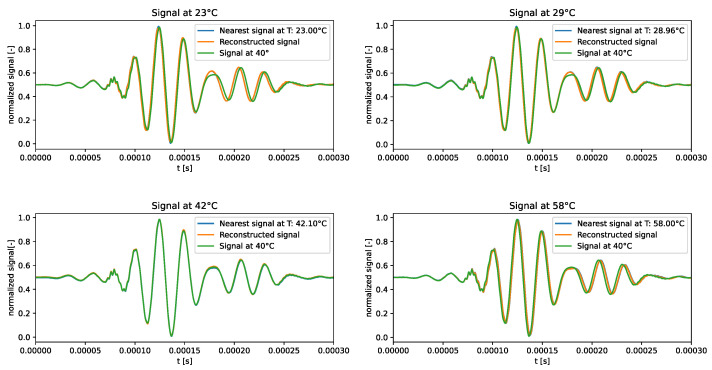
Test signals generated by the f-VAE trained over the standard dataset.

**Figure 11 sensors-24-01494-f011:**
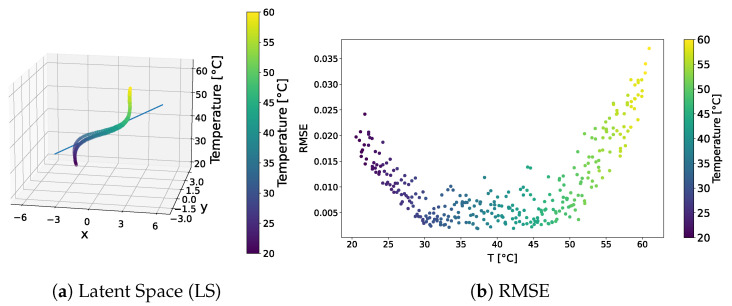
f-VAE—band dataset. Latent-space distribution and RMSE.

**Figure 12 sensors-24-01494-f012:**
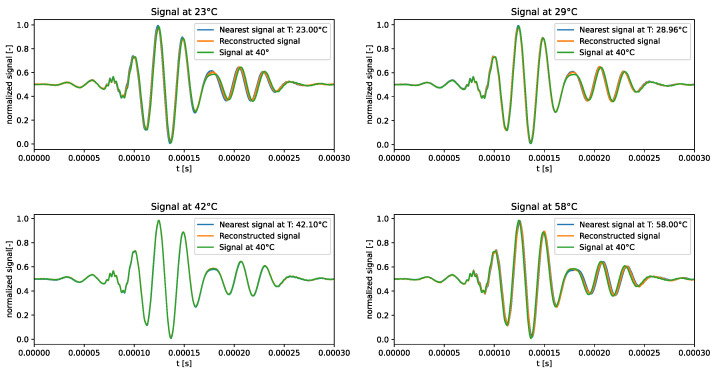
Test signals generated by the f-VAE trained over the band dataset.

**Figure 13 sensors-24-01494-f013:**
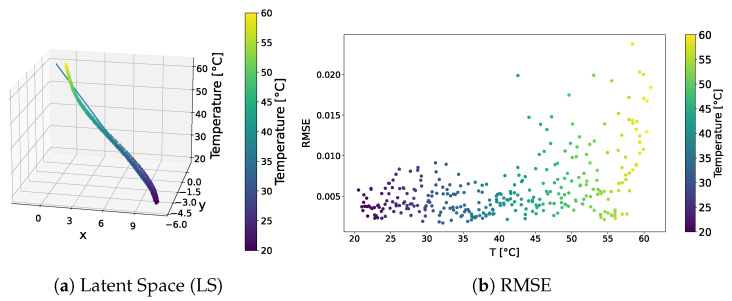
f-VAE—sparse dataset. Latent-space distribution and RMSE.

**Figure 14 sensors-24-01494-f014:**
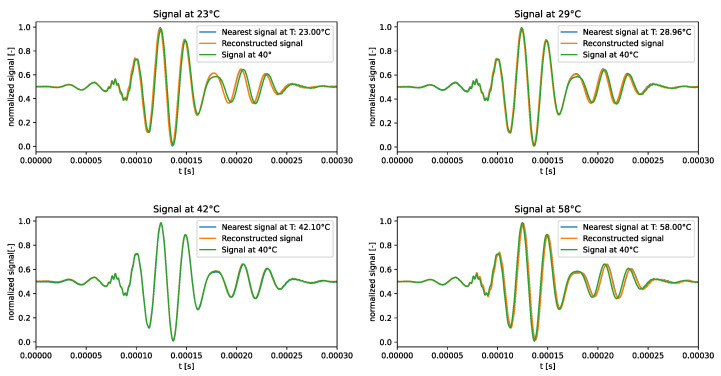
Test signals generated by the f-VAE trained over the sparse dataset.

**Table 1 sensors-24-01494-t001:** Summary of the neural network architecture.

Layer	Number of Neurons	Activation Function	
Input	1 × 13,108	-	**Optimizer:** “Adam”
Dense	128	Sigmoid Linear Units	
Dense	64	Sigmoid Linear Units	
Dense	16	Sigmoid Linear Units	
Latent Space	2	-	
Sampling	1	-	
Dense	16	Sigmoid Linear Units	
Dense	64	Sigmoid Linear Units	
Dense	128	Sigmoid Linear Units	
Output	1 × 13,108	Sigmoid	

## Data Availability

The codes presented in the manuscript are available in GitHub: https://github.com/lorenzomie/VAE-generative-temperature-signal-for-CFRP-plate (accessed on 21 February 2024).

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
