# Peer review of "Mitigating the Impact of Temperature Variations on Ultrasonic Guided Wave-Based Structural Health Monitoring through Variational Autoencoders"

_sensors, 2024, doi:10.3390/s24051494_

Round 1

Reviewer 1 Report

Comments and Suggestions for Authors

This manuscript focusses on the mitigation of the impact of temperature variation on ultrasonic guided wave-based structural health monitoring trough autoencoders. The paper is well written but some aspect must be clarified. My major observations are the following:

1) It is not clear how the measurements are sampled, the authors write: "Signals were sampled over a frequency band from 40 KHz to 260 KHz, with intervals of 20KHz". I cannot figure out what is the sampling frequency of the piezoelectric sensors and in the considered dataset what is the sampling frequency adopted.

2) It is not clear if a the datasets have been partitioned in training and test sets, it is always recommended to evaluate performance on a dataset not used to train the algorithm to avoid generalization error. Please highlight also the percentage of samples used for training and the one used for test.

3) SiLU activation function has been introduced in table 1 but the acronym is not defined.

4) It is not clear to me if the authors used raw time series data from the 12 sensors or exploited spectral information, if raw time series data has been used, how the measurements have been sincronized?

5) I do not understand why the author used a Sigmoid function in the output layer of the autoencoder when it is common choice to use Linear activation function. The goal of the autoencoder is to reconstruct the input signal in the output layer, that means that the output should exhibit the same activation of the input. Using a Sigmoid activation function the output is forced to assume values 0 or 1 that makes much more difficult to mimic the input continuous activations. 

Comments on the Quality of English Language

The paper is well written, only minor editing of english is required.

Author Response

Thank you for your comments. The manuscript was modified according to the concerns pointed out in the list. All the modifications are written in red in the updated manuscript. Below you can find our point-by-point response to your comments.

  1. The description was revised to clarify how signals were sampled:
    Signals were acquired in a round-robin fashion including measurements from 40kHz to 260kHz, with step 20kHz.  
  2. The reviewer is right, training and testing are usually performed on different datasets. In our work, due to the unsupervised nature of the proposed method, training is performed using all or some (depending on the case study we consider, i.e., standard, sparse or band dataset) of the acquired signals. Then, testing is performed by generating new signals at temperature that was not seen during training. That is, the generated signals are always out of the training dataset. This was better clarified in the manuscript by adding the following sentence before listing the error metrics:
    As the method operates in an unsupervised manner, temperature labels were not supplied during training. Moreover, testing was performed by generating signals at temperatures not considered during training, so to verify the generalization capabilities of the proposed method.
  3. Thank you for highlightinh the missing acronym. SiLU was replaced with "Sigmoid Linear Units" in Table 1.
  4. The authors only utilized raw time series data in the model. Signals had already been synchronized during acquisition, which was not perfomed in this work. The reviewer is referred to the following papers to get more details about experimental operations (which were not conducted by the authors):

    Moll, J.; Kexel, C.; Pötzsch, S.; Rennoch, M.; Herrmann, A. S.: Temperature affected guided wave propagation in a composite plate complementing the Open Guided Waves Platform, Scientific Data, 2019, 6:191, DOI: 10.1038/s41597-019-0208-1

    Moll, J.; Kexel, C.; Pötzsch, S.; Rennoch, M.; Herrmann, A. S.: Temperature affected guided wave propagation in a composite plate complementing the Open Guided Waves Platform, 2019, figshare, collection, DOI: 10.6084/m9.figshare.c.4488089.v1

  5. The unconventional choice of using the Sigmoid function instead of the Linear activation function does not limit the capabilities of the VAE. In fact, since the input to the network is linearized between 0 and 1, the reconstructed input in the output layer should also have values in the range 0 to 1. The Sigmoid function provides continuous values in that range, and the authors verified that the performance of the method is not affected at all by replacing the Sigmoid function with the Linear function.

Reviewer 2 Report

Comments and Suggestions for Authors

The author studied the effect of temperature on ultrasonic guided waves and its predictive reconstruction methods. Specific suggestions are as follows.

1. The introduction did not provide sufficient discussion of relevant literature, such as the citation of 1-10 lacking direct meaning.

From a data perspective, there seems to be little difference between data at different temperatures. There may be significant effects at ultra-high or ultra-low temperatures, but this article did not provide relevant research.

3. In terms of application value, the tomography algorithms targeted in this article mainly focus on calculating the amplitude of the signal, and do not pay much attention to phase and other information.

4. The entire article does not meet the requirements of sensors for a 16 page article.

Author Response

The authors thank the reviewer for his comments on the paper. Below are the answers to his comments:

  1. The Introduction was improved. For the sake of brevity, the new paragraphs are not reported here. 
  2. The reviewer is right, temperature has a limited influence on the signals. This is exaclty the point of the paper, which focuses on a method that is accurate enough to also capture such little variations. Moreover, note that damage also has a similar influence on signals, and this makes it hard to differentiate between the influence of temperature and damage on the diagnostic signals.
    No ultra-high or ultra-low temperature was considered because the experimental campaing was limited to the temperature range written in the paper. Making that range wider is out of the scope of this contribution. 
  3. We do not fully understand this comment. No tomographic algorithm was presented in the manuscript, but only a generative AI method for dealing with Lamb waves at different temperatures. This tool will be implemented in the future in a tomographic algorithm that can localize damage by considering the whole time history. Hence, the tomographic algorithm (which is NOT presented in this work) will deal with amplitude, phase and other information carried by the diagnostic signal.
    Moreover, note that the AI tool can accurately predict amplitude variations and phase shifts due to temperature variations.
  4. The Instructions for Authors say "The recommended length of an Article is more than 16 journal pages". This is only the recommended length, and several papers with less that 16 pages can be found in the journal (even with less than 14 pages).

Reviewer 3 Report

Comments and Suggestions for Authors

This manuscript presents an artificial intelligence based framework to mitigate the impact of temperature variations on guided ultrasonic wave based structural health monitoring. This study sets itself apart from recent previous studies that have used autoencoders to obtain training data at various temperatures by applying variable autoencoders to predict guided ultrasonic wave signals traveling through a composite panel at temperatures not present in its training data set. The methods, results, and discussion are presented in a clear fashion. This work will be of interest to the ultrasonics and nondestructive evaluation communities. The AI-based sensing approach, which mitigates environmental challenges, is well aligned with the scope of the journal. I therefore recommend its publication in Sensors. 

I have a few minor comments that I would like the authors to address:

1. NDE based on ultrasonic guided waves is specific to plate and shell-like structures. This should be mentioned in the first paragraph of the Introduction.

2. The temperatures tested for the training data set by Abbasi et al (in reference 20) should be mentioned.

3. Typographical error - Introduction should have upper case I.

4. Please provide more details of the CFRP plate on which the signals were acquired such as the thickness, ply orientation, geometry/ dimensions, etc. I presume this information is given in OpenGuidedWaves platform, but will be good to repeat here for completeness.

5. Plate thickness is particularly important in the context of the frequency of the guided ultrasonic waves samples (40 - 260 kHz).

6. What is the rationale for selecting a small varying temperature range (20 - 60C). How would the dataset (and subsequent artificial intelligence models) be affected if this data set was collected over larger varying temperature range?

7. Please comment on the scalability of the proposed framework? What is the computational expense to compute reconstructed signals? How will this scale up for large structures that require hundreds (or more) sensor channels?

8. The axes labels in Figure 3, 5, 7, 9 and 11 need to be in larger font.

Author Response

Thank you for your comments. Below you can find the point-by-point response to the suggestions. All the modifications were written in red in the revised manuscript.

  1. The reviewer is right. The first paragraph of the introduction was modified as follows:
    This method is specific to plate and shell-like structures and employs a single piezoelectric transducer as an actuator, transmitting ultrasonic waves into the material, while multiple strategically positioned transducers serve as receivers to capture the transmitted waves.
  2. The information was added to the introduction:
    In Abbassi et. al [], autoencoders demonstrated the capability to detect damage at different positions independently of temperature, which varied from 20°C to 60°C. The training dataset encompassed data from all tested temperatures.
  3. Fixed, thank you for pointing out.
  4. Such information was missing in the manuscript. We modified the first paragraph of Section 2.1 by adding the following details:
    The dataset utilized in this study was sourced from the OpenGuidedWaves [24] platform, a repository offering comprehensive datasets of wave signals acquired on a carbon fiber-reinforced polymer (CFRP) plate which has a quasi-isotropic layup with a stacking sequence of [45/0/−45/90/−45/0/45/90]S. The plate's dimensions measure 500mm × 500mm, and it has a thickness of 2mm.
  5. Indeed, thickness is important. This information was added to Section 2.1.
  6.  We selected the temperature range of 20 - 60°C based on the available data in the dataset. All available temperatures within this range were utilized in our study. While a larger temperature range would increase the training time and the size of the dataset, incorporating more temperatures would enhance the model's ability to provide insights across a broader spectrum of varying environmental conditions. Furthermore, with more variability in the dataset, the VAE architecture may need to be expanded to allow better adapting to bigger data. The following paragraph was added to the end of the section "Dicussion":
    Signal generation involved negligible computational time, in the order of milliseconds, despite training taking a few minutes on a hexacore Intel i7 - 10850 CPU @ 2.70 GHz. Scaling up to larger structures with numerous sensors is expected to lead to longer training times, since the network architecture would need to be enlarged for better capturing signals variability. However, the time for generating new signals is expected not to be affected and to remain in the order of milliseconds. The same considerations apply to training over signals acquired over a wider temperature range.
  7. We appreciate the observation. The scalability, computational expenses, and considerations for large structures have been addressed in the chapter Discussion. Have a look at the answer to question 6.
  8. The font size was increased in all the figures mentioned by the reviewer.

Round 2

Reviewer 1 Report

Comments and Suggestions for Authors

The authors reply to all the reviewer questions.

Author Response

Thank you for your feedback.

No further modifications are required according to this Reviewer.

Reviewer 2 Report

Comments and Suggestions for Authors
The author claims that ‘No tomographic algorithm was presented in the manuscript, but only a generative AI method for dealing with Lamb waves at different temperatures.’ However, in the abstract, the author emphasizes that ‘To this purpose, the performance of traditional methods based on tomographic algorithms has been overcome by machine learning approaches, which allow processing a larger amount of data without losing diagnostic information’.
The motivation of this article is not clear. At different temperatures, damage detection or localization is usually carried out through amplitude, phase, or time difference. There is no need to reconstruct the signal based on temperature changes in the method.
Moreover, temperature can alter the properties of materials, such as density, Poisson's ratio, etc. The effect of temperature on ultrasound should be numerically solved by introducing changes in material parameters. For example, in solving the Lamb wave dispersion curve of a plate-like structure.

Author Response

We understand the Reviewer's concerns. Our claim that no tomographic algorithm was presented in the manuscript is correct. In fact, we intend to present a generative AI-based tool that can be used to enhance the performance of tomographic algorithms, and NOT to develop a new algorithm for damage localization by scratch. This was already clearly stated in the conclusions through the following paragraph:

"Future work will focus on implementing forced variational autoencoders and singular value decomposition into unsupervised frameworks for damage detection, localization and quantification. This will allow making a step towards robust structural health monitoring tools that are not influenced by varying environmental and operational conditions".

Still, we believe that mentioning tomographic algorithms for damage localization and their limitations is fundamental to introduce the motivation behind this study. Otherwise, researchers not dealing with ultrasonic guided waves measurements could find it hard to understand the reason why neutralizing temperature effects would improve the damage diagnosis performance.

Based on the motivations above, and according to the Reviewer's suggestions, we modified the abstract (the modifications are highlighted in red) to make the aim and motivation of the paper clearer:

"Structural health monitoring (SHM) has become paramount for developing cheaper and more reliable maintenance policies. The advantages coming from adopting such process have turned out to be particularly evident when dealing with plated structures. In this context, state-of-the-art methods are based on exciting and acquiring ultrasonic guided waves through a permanently installed sensor network. A baseline is registered when the structure is healthy, and newly acquired signals are compared to it to detect, localize and quantify damage. To this purpose, the performance of traditional methods has been overcome by data-driven approaches, which allow processing a larger amount of data without losing diagnostic information. However, to date, no diagnostic method can deal with varying environmental and operational conditions (EOCs). This work aims to present a proof-of-concept that state-of-the-art machine learning methods can be used for reducing the impact of EOCs on the performance of damage diagnosis methods. Generative artificial intelligence was leveraged to mitigate the impact of temperature variations on ultrasonic guided wave-based SHM. Specifically, variational autoencoders and singular value decomposition were combined to learn the influence of temperature on guided waves. After training, the generative part of the algorithm was used to reconstruct signals at new unseen temperatures. Moreover, a refined version of the algorithm called forced variational autoencoder was introduced to further improve the reconstruction capabilities. The accuracy of the proposed framework was demonstrated against real measurements on a composite plate."

Furthermore, the following sentence was added in the Section "Introduction": 

"The aim of this work is to present a proof-of-concept that generative AI can be used to neutralize the effect of EOCs on the damage diagnosis task. In fact, the proposed tool can be used to reconstruct signals under varying EOCs, thus enhancing the damage fingerprint and helping damage diagnosis algorithms reduce false alarms and improve the quality of predictions."

Instead, we diasgree on the lack of need to reconstruct signals at different temperatures. In fact, dealing with EOCs is a long lasting problem in the field of SHM, and no definitive method to overcome such limitation has been proposed in the literature so far. A huge amount of papers has been written about the topic, and we believe there is no need to make the problem clearer to researchers. We also strongly believe that proposing a proof-of-concept that state-of-the-art generative AI methods can be leveraged to reconstruct signals under varying EOCs can be of interest to the research community. That is, deploying a tool that can be integrated into traditional and/or data-driven methods to reconstruct signals at any given EOCs would be beneficial to damage diagnosis algorithms, since the damage fingerprint would be strongly highlighted by such approach.

Finally, we do not understand the Reviewer's concerns about numerical simulations. The case study is FULLY EXPERIMENTAL and NO NUMERICAL SIMULATIONS WERE CONSIDERED. Moreover, it is well known by the research community that introducing the complexities that the Reviewer mentions would make the simulations prohibitively expensive. This would be incompatible with the usage of state-of-the-art data-driven solutions, since a database large enough for training the algorithms could not be generated. Furthermore, the solution we propose DOES NOT REQUIRE NUMERICAL SIMULATIONS, since variational autoencoders are fully trained on the acquired baseline, which can be experimental (as also done in our work).